# The Involvement of Endogenous Enkephalins in Glucose Homeostasis [note 1]

**DOI:** 10.3390/biomedicines11030671

**Published:** 2023-02-22

**Authors:** Vanessa Escolero, Laica Tolentino, Abdul Bari Muhammad, Abdul Hamid, Kabirullah Lutfy

**Affiliations:** 1Graduate College of Biomedical Sciences, Western University of Health Sciences, Pomona, CA 91766, USA; 2Burrell College of Osteopathic Medicine, 3501 Arrowhead Drive, Las Cruces, NM 88001, USA; 3Cardiovascular and Metabolism (CVM), Janssen Research and Development, 100 Binney Street, 5th Floor, Cambridge, MA 02141, USA; 4Department of Pharmaceutical Sciences, College of Pharmacy, Western University of Health Sciences, Pomona, CA 91766, USA

**Keywords:** enkephalins, high-fat diet (HFD), obesity/diabetes type 2, oral glucose tolerance test (GTT), insulin tolerance test (ITT)

## Abstract

Obesity has nearly tripled since 1975 and is predicted to continue to escalate. The surge in obesity is expected to increase the risk of diabetes type 2, hypertension, coronary artery disease, and stroke. Therefore, it is essential to better understand the mechanisms that regulate energy and glucose homeostasis. The opioid system is implicated in regulating both aspects (hedonic and homeostatic) of food intake. Specifically, in the present study, we investigated the role of endogenous enkephalins in changes in food intake and glucose homeostasis. We used preproenkephalin (ppENK) knockout mice and their wildtype littermates/controls to assess changes in body weight, food intake, and plasma glucose levels when mice were fed a high-fat diet for 16 weeks. Body weight and food intake were measured every week (n = 21–23 mice per genotype), and at the end of the 16-week exposure period, mice were tested using the oral glucose tolerance test (OGTT, n = 9 mice per genotype) and insulin tolerance test (n = 5 mice per genotype). Our results revealed no difference in body weight or food intake between mice of the two genotypes. However, HFD-exposed enkephalin-deficient mice demonstrated impaired OGTT associated with reduced insulin sensitivity compared to their wildtype controls. The impaired insulin sensitivity is possibly due to the development of peripheral insulin resistance. Our results reveal a potential role of enkephalins in the regulation of glucose homeostasis and in the pathophysiology of diabetes type 2.

## 1. Introduction

Worldwide obesity has nearly tripled since 1975, and current trends are expected to continue to rise [1]. As per World Health Organization (WHO) 2022 statistics, more than 1 billion people are obese worldwide [2]. In addition, a recent report from the Center for Disease Control (CDC) estimated that the prevalence of obesity from 2017 to 2020 in the United States was 41.9% among adults and 19.7% among children and adolescents [3]. The increased rates of obesity are driven primarily by the overconsumption of energy-dense foods and an increase in sedentary lifestyles, resulting in a shift in energy balance that leads to excessive fat accumulation. Furthermore, this increases the risks for chronic comorbidities such as diabetes type 2, coronary artery disease, hypertension, stroke, and certain cancers [4]. Given the continued rise in obesity and diabetes type 2 over the past decades, understanding the mechanisms that regulate energy homeostasis and glucose metabolism would be beneficial in developing novel therapeutics to prevent or at least reduce the devastating public health and socioeconomic consequences of obesity and diabetes type 2.

The endogenous opioid system, consisting of opioid peptides (beta-endorphin, dynorphin, and enkephalins) and their corresponding G-protein-coupled receptors, namely, delta (DOP), kappa (KOP), and mu (MOP) opioid receptors [5,6], is heavily implicated in feeding behavior. These opioid peptides and their receptors are highly expressed in brain regions involved in the rewarding and hedonic aspects of feeding, such as the mesolimbic dopaminergic system. In addition, these opioid peptides are localized within the hypothalamus and, thus, are likely involved in modulating homeostatic signals received from the peripheral tissues [7]. Pharmacological studies have shown that opioid receptor agonists stimulate food intake [8,9], while opioid receptor antagonists suppress food intake [8,10,11]. However, the role of each opioid peptide in feeding behavior and, notably, glucose homeostasis is not fully understood.

Previous studies have shown that the administration of enkephalin and enkephalin analogs increases the intake of palatable food [12,13]. Moreover, an increase in enkephalin mRNA was observed in the brain of rats consuming a high-fat diet, reinforcing a positive relationship between a high-fat diet and enkephalin [13]. In addition, a 2015 study showed that mice lacking the ability to produce enkephalin gained less weight when compared to their wildtype controls despite equal amounts of food intake [14].

Enkephalins are found in several peripheral organs, including the pancreas, liver, adipose, skeletal muscle, lungs, heart, as well as in the central and peripheral nervous systems, suggesting the involvement of enkephalins in various biological processes such as feeding behavior [15]. Indeed, enkephalins have been shown to play a functional role in glucose metabolism. For example, one of the first studies conducted in the 1980s demonstrated the presence of opioid receptors in the islets and a low dose of enkephalin-stimulated insulin secretion in vitro [16]. In contrast, in vitro high concentrations of enkephalin inhibited insulin release [16,17], raising the possibility that enkephalins may regulate insulin secretion and glucose homeostasis. Interestingly, delta-opioid-receptor-deficient mice fed a chow diet demonstrated improved glucose tolerance compared to wildtype; however, when challenged with a high-fat diet, the difference between genotypes disappeared [18]. Most recently, enkephalins have been implicated in the beiging process of white adipocytes through the activation of group 2 innate lymphoid cells [19]. However, the role of enkephalins is partially characterized in feeding and glucose homeostasis. Therefore, our study aims to further understand the role of the opioid peptides, enkephalins, in the regulation of body weight and glucose homeostasis.

## 2. Materials and Methods

### 2.1. Animals

We utilized enkephalin knockout mice that were previously generated by Konig and colleagues [20]. Mice lacking the preproenkephalin (ppENK) gene and their wildtype controls were fully backcrossed on a C57BL/6 mouse strain and bred in-house. Pups were weaned on day 21 and genotyped according to our earlier reports [21,22]. In total, there were 21 wildtype and 19 knockout male mice used, but 2 wildtype and 2 knockout mice were sacrificed early in the experiment due to severe dermatitis and excluded from the data analysis. Mice were housed in recyclable plastic cages with bedding and ad libitum access to food and water under a 12 h light/12 h dark cycle in a temperature- and humidity-controlled room. Animal cages were changed to new cages every two weeks. All animal experiments were performed in accordance with the NIH guideline for the care and use of animals in research and approved by the Western University of Health Sciences Institutional Animal Care and Use Committee (Pomona, CA, USA), protocol number R20IACUC024.

### 2.2. Diet

Mice were maintained on a regular laboratory chow diet (Envigo Teklad Global Diets, 2018, 18% protein rodent diet; 24% kcal protein, 18% kcal fat, and 58% kcal carbohydrates; 3.1 kcal/g) for one week. Then, for the remainder of the study, mice were placed on a high-fat diet obtained from Research Diets, Inc (60% kcal fat, 20% kcal carbohydrates, and 20% kcal protein with blue dye: 5.21 kcal/g formulation D12492).

### 2.3. The Role of Enkephalins in Body Weight and Food Intake

To determine the involvement of endogenous enkephalins in body weight and food intake, we used mice lacking enkephalins and their wildtype littermates/controls that we described previously [21,22]. The experiment was conducted over the course of 17 weeks. Each animal had their body weight and food intake measured in grams (g), once weekly. Initially, animals were isolated and housed one per cage and given a regular diet for the first week to habituate to a single mouse per cage condition. This was performed to enable us to measure the food intake of each mouse accurately and also to avoid fighting in the cage, as this can be stressful and affect food intake and importantly blood glucose. At week 2, they were switched to the high-fat diet (HFD) and remained on the HFD through week 17. The experiment was repeated four times with 4–6 mice per cohort (n = 21–23 mice per genotype).

### 2.4. The Role of Enkephalins in Glucose Homeostasis

To determine the role of enkephalin in glucose homeostasis, an oral glucose tolerance test (OGTT) was conducted at week 17 in 2 different sets (n = 3–5 mice per cohort) of mice (n = 8 mice per genotype). The protocol was according to our earlier report [23]. Mice were fasted overnight starting at 10 p.m., for a total of 11 h with ad libitum access to water during this period. The next morning (9 AM), while the animals were still kept fasted, a small piece of the tail was snipped, and the baseline blood glucose level was measured for each mouse, using a glucometer (Embrace Omnis Blood Glucose Monitoring System/Omnis Health Embrace Blood Glucose Test Strip). Mice were then weighed and administered a 2 g/kg of glucose solution via oral gavage. Blood glucose levels were then measured at 30, 60, 90, 120 min after the glucose challenge. At the end of the experiment, mice were restored with ad libitum access to water and food.

### 2.5. The Role of Enkephalins in Insulin Sensitivity

To determine the role of enkephalins in insulin sensitivity and provide complementary results to the glucose tolerance test, a separate set of mice underwent an intraperitoneal insulin tolerance test (ITT) at week 17. The ITT protocol was according to our earlier report, except we fasted the mice for 6 h to avoid hypoglycemia [23]. Briefly, mice were kept fasted for a total of 6 h with access to ad libitum water. Mice were weighed and then given 0.75 U/kg insulin intraperitoneally (Humulin R 100 U/mL, Eli Lilly and Company). Tail vein blood was used to measure blood glucose levels using a glucometer, as described above. Blood glucose levels were measured before (time 0) and at 15, 30, 45, 60, 90 min after insulin injection. At the end of the experiment, mice were restored with ad libitum access to water and food.

### 2.6. The Role of Enkephalin in Plasma Insulin Levels

Considering changes in plasma insulin level can alter blood glucose levels, we also assessed if the lack of enkephalins would alter plasma insulin levels. Briefly, mice (n = 8–10 mice per genotype) were euthanatized through decapitation. Whole trunk blood was collected in ethylenediaminetetraacetic acid (EDTA)-containing tubes and centrifuged at 14,000 rpm for 10 min at 4 °C. Plasma samples were collected and stored in a −80 °C freezer until analysis. Mouse insulin levels were measured with a commercially available ELISA kit (Rat/Mouse Insulin ELISA Kit, Cat. # EZRMI-13K). The protocol supplied by the kit manufacturer was followed. Briefly, a 96-well plate pre-coated with monoclonal mouse anti-rat insulin antibodies was loaded with six rat insulin standards, two quality controls, background, and samples. Each sample was diluted 2-fold with the assay buffer according to the manufacturer’s instructions and run in duplicates. The absorbance was read at a wavelength of 450 nm with a plate reader. Based on the standard curves run in duplicate on each plate, plasma insulin concentration was determined with linear regression using GraphPad Prism 9.1.2 software (San Diego, CA, USA).

### 2.7. The Role of Enkephalin in Plasma Leptin Levels

Given that changes in leptin levels can affect plasma glucose levels, we also examined if the lack of enkephalins would alter plasma leptin levels. Mice (n = 9–11 per genotype) were euthanized, and trunk blood was corrected and prepared, as described above. Mouse leptin levels were measured with a commercially available ELISA kit (Mouse Leptin ELISA Kit, Cat. # EZML-82K). The protocol supplied by the kit manufacturer was followed. Briefly, a 96-well plate pre-coated with pre-tittered antiserum was loaded with six rat insulin standards, two quality controls, background, and samples. Each sample was analyzed in duplicates and diluted 2-fold with assay buffer according to the manufacturer’s instructions. The absorbance was read at a wavelength of 450 nm with a plate reader. Based on the standard curves run in duplicate on each plate, plasma leptin concentration was determined with linear regression using GraphPad Prism 9.1.2 software (San Diego, CA, USA).

### 2.8. Statistical Analysis

Data were expressed as mean ± SEM. Time-course data were analyzed using two-way repeated measures analysis of variance (ANOVA) followed by Fisher’s LSD post hoc multiple comparisons. The area under the curve (AUC) for the OGTT and ITT time-course data was calculated using the trapezoidal rule. Plasma insulin and leptin levels were measured using linear regression analysis. A *p*-value < 0.05 was considered significant. Statistical analysis and graphing were completed in GraphPad Prism version 9 (San Diego, CA, USA).

## 3. Results

### 3.1. Body Wight Measurement in Wildtype and Knockout Mice

We first determined if endogenous enkephalins regulate body weight. Figure 1 shows the raw body weight of male wildtype (ppENK+/+) and knockout (ppENK-/-) animals for the duration of the experiment. The knockout mice displayed a similar pattern in body weight gain as the wildtype mice. A mixed-effects analysis revealed a significant effect for time (*p* < 0.0001) but no effect for genotype (*p* = 0.9673) or genotype x time (*p* = 0.9946). The percentage change in body weight was also not different between the wildtype and knockout animals (Figure 1b). A mixed-effects analysis revealed a significant effect for time (*p* < 0.0001) but not for genotype (*p* = 0.0136) or genotype x time (*p* = 0.1714). This result reveals that endogenous enkephalins may not play a significant role in HFD-induced weight gain.

### 3.2. Food Intake Measurement in Wildtype and Knockout Mice

To assess if endogenous enkephalins alter food intake, we used mice lacking enkephalins and their wildtype littermates. Figure 2 shows the daily food intake in grams (g) between wildtype and knockout animals fed the regular and HFD. Week 1 demonstrates the initial measurement of the regular diet (g) followed by the HFD on week 2, as marked on the graph. As can be observed, there was a drop in daily food intake from week 1 to week 2, likely due to the HFD containing higher total calories than the regular diet, leading to reduced consumption. However, there were no significant differences in food intake between wildtype and knockout mice. A mixed-effects analysis revealed a significant effect for time (*p* < 0.0001) but no effect for genotype (*p* = 1.023) or genotype x time (*p* = 0.6778). This result suggests that endogenous enkephalins may not regulate regular diet or HFD consumption.

### 3.3. Oral Glucose Tolerance Test

To examine the role of enkephalin in glucose homeostasis, an oral glucose tolerance test (OGTT) was conducted in mice lacking enkephalins and their wildtype littermates. The study was repeated twice with n = 4–5 per cohort (n = 9 mice per genotype). Briefly, the mice were fasted overnight, and fasted blood glucose levels were measured before administering a glucose solution via oral gavage. Subsequently, a 2 g/kg oral glucose gavage was administered, and blood glucose levels (mg/dL) were measured between wildtype and knockout mice at specific time points for a total of 90 min, as shown in Figure 3. We observed glucose intolerance in ppENK knockout mice throughout all time points after oral gavage compared to wildtype mice. A two-way ANOVA analysis revealed a significant effect for time (F(3, 48) = 57.42, *p* < 0.0001), genotype (F(1, 16) = 7.106, *p* = 0.0169), but no interaction between time x genotype (F(3, 48) = 1.013, *p* = 0.3953). Fisher’s LSD post hoc test revealed a significant increase at 30 and 60 min in mice lacking enkephalin compared to the wildtype. An unpaired student’s *t*-test of the AUC also showed a significant (*p* < 0.02) increase in blood glucose levels in mice lacking enkephalins compared to their wildtype controls (t = 2.84, df = 16; Figure 3b). This rise could be due to changes in basal fasting glucose between wildtype and knockout mice. While repeated measures ANOVA did not reveal any significant difference in basal fasting glucose between mice of the two genotypes, this was a significant difference when basal levels were analyzed by unpaired student’s *t*-test (t = 2.41; df = 16; *p* < 0.02). However, this difference was not observed when animals were fasted for 6 h for the ITT (see below). These results suggest that glucose handling in the absence of enkephalins is reduced. However, basal fasting glucose is only affected when the duration of fasting is 12 h.

### 3.4. Insulin Tolerance Test

We conducted an insulin tolerance test to determine whether the impaired glucose tolerance in the knockout than wildtype mice was due to reduced sensitivity to insulin. In total there were seven ppENK wildtype (ppENK+/+) and five knockout (ppENK-/-) male mice; however, two wildtype mice were excluded from the analysis, as these mice were non-responsive to the insulin challenge. The blood glucose levels of these mice remained elevated throughout the experiment compared to the rest. Briefly, mice were kept fasted for 6 h, and fasting blood glucose levels were measured before the intraperitoneal insulin injection. Following a 0.75 U/Kg insulin injection was administered intraperitoneally, blood glucose levels (mg/dL) were measured and compared between wildtype and knockout for 90 min, as shown in Figure 4. We observed knockout mice were glucose-intolerant at all time points following the insulin injection. These findings were consistent with the OGTT results, demonstrating that ppENK-deficient mice have reduced glucose tolerance and are insulin resistant. A two-way ANOVA analysis demonstrated a significant effect of time (F(5, 40) = 11.70, *p* < 0.0001), genotype (F(1, 8) = 11.08, *p* = 0.0104), but no interaction between time x genotype (F(5, 40) = 1.090, *p* = 0.3806). The Fisher’s LSD test revealed that there was a significant increase in blood glucose at 45 and 90 min in knockout mice compared to their wildtype controls. Analyses of the AUC also revealed a significant (*p* < 0.01) increase in blood glucose in mice lacking enkephalins than their wildtype littermates (t = 3.60; df = 8; Figure 4b). These results suggest that insulin sensitivity is impaired in mice lacking enkephalins following exposure to HFD for 16 weeks.

### 3.5. Insulin and Leptin Measurement in Wildtype and Knockout Mice

We then investigated if enkephalins could also regulate the level of insulin or leptin, which could explain the changes observed in the OGTT and ITT. We measured the plasma insulin and leptin levels through enzyme-linked immunosorbent assay (ELISA) using a commercially available ELISA kit in wildtype and knockout mice. As shown in Figure 5a, the plasma insulin levels were comparable in wildtype and knockout mice. An unpaired student’s *t*-test was utilized for statistical analysis, which revealed no significance (t(df = 16) = 0.3155, *p* = 0.7565). In addition, plasma leptin levels were also measured utilizing a commercially available ELISA kit. As shown in Figure 5b, there were no significant differences in plasma leptin between wildtype and knockout mice. For statistical analysis, an unpaired student’s *t*-test was used, which demonstrated no significance (t(df = 16) = 0.2194, *p* = 0.8288). This result shows that enkephalins may not regulate plasma insulin or leptin levels in mice exposed to HFD for 16 weeks.

## 4. Discussion

The regulation of nutrient intake and metabolism is tightly regulated by the interplay of numerous systems [24,25]. In this study, we focused on the opioid system, specifically the role of enkephalins in food intake and glucose homeostasis. As stated earlier, only a handful of studies have examined the role of enkephalins in energy and glucose homeostasis. The findings in this study are consistent with the literature, demonstrating that opioids play a role in glucose and energy homeostasis; specifically, our study suggests that enkephalins are likely involved in glucose homeostasis.

Our results suggest that the deletion of enkephalins does not affect food intake or body weight, as there was no statistically significant difference in raw body weight or the percentage of body weight changes between mice lacking enkephalins and their wildtype controls. Our results contrast that of Mendez et al. (2015), which was one of the first studies that demonstrated mice lacking enkephalins were resistant to diet-induced obesity compared to wildtype controls, although there was no difference in food consumption between mice of the two genotypes [14]. Enkephalins are implicated in the beiging of white adipocytes through the activation of uncoupling protein-1 (UCP-1) [19]. However, the different diets and the duration of exposure may explain the discrepancy between our results and that of Mendez and colleagues who utilized a “cafeteria diet” composed of snacks, such as cheese puffs, pretzels, cereal, marshmallows, pepperoni, etc. [14]. Age differences should also be investigated as the starting age of animals for our study ranged from 10 to 26 weeks compared to 21 weeks in the former study [14].

Previous studies have shown differences in plasma glucose levels in mice lacking kappa (KOP), mu (MOP), and delta (DOP) compared to their respective wildtype controls [18,26,27]. The current literature demonstrates that MOP knockout mice have enhanced glucose tolerance when fed a high-fat diet, and similar results have been observed in mice lacking DOP or KOP [26,27,28]. However, the role of each opioid peptide in regulating glucose homeostasis is unclear. Thus, we used mice lacking enkephalins and their wildtype littermates/age-matched controls to assess whether basal or oral glucose tolerance tests would be altered in the absence of enkephalins. We also assessed insulin tolerance test in mice lacking enkephalins and their wildtype littermates/controls to assess if the change in plasma glucose observed in the OGGT is a result of changes in insulin sensitivity. Our data show elevated basal blood glucose levels in mice of both genotypes, which is expected based on the literature and consistent with insulin resistance in mice fed a high-fat diet [29,30]. However, enkephalin knockout mice had higher blood glucose in the OGTT and impaired insulin sensitivity in the ITT compared to their wildtype littermates/controls. These findings suggest that enkephalins may offer protection from the negative effects of a dense, high-fat diet on glucose homeostasis and insulin sensitivity in mice. While the 12 h fasting led to a significant difference in basal glucose levels between wildtype and knockout mice, the 6 h fasting was without any effect on basal glucose level (ITT data). It appears that longer fasting may be necessary to yield a difference in basal fasting glucose levels between wildtype and knockout mice, suggesting that enkephalins may regulate basal glucose levels in mice exposed to HFD for 16 weeks and exposed to 12 h of fasting. On the contrary, enkephalin knockout mice fed a chow diet did not show an impaired glucose tolerance test (Appendix A), which is consistent with data obtained in DOP knockout mice fed a regular diet, as they did not show an impaired glucose tolerance test compared to their high-fat-diet-fed wildtype counterparts [18].

In addition, our enkephalin knockout mice demonstrated impaired insulin sensitivity during the insulin tolerance test, indicating elevated glucose levels are possibly due to the development of peripheral insulin resistance. It needs to be noted that the rise in glucose could also be due to reduced insulin release from the pancreas. However, the measured plasma insulin levels showed no significant difference between mice of the two genotypes. Additionally, no change in plasma leptin was found between wildtype and knockout mice, but further studies are needed to comprehensively assess the role of enkephalins in plasma insulin and leptin regulation. At the time of blood collection, our mice were not fasted, which likely contributed to the measured results. Additionally, the mice were not challenged with glucose, which allowed us to measure glucose-stimulated insulin secretion. Therefore, repeating this experiment using glucose-stimulated insulin secretion (GSIS) will provide useful information concerning the role of enkephalins in regulating insulin secretion. Regardless, the OGTT and ITT do indicate that these mice are insulin-resistant.

Enkephalins are widely distributed throughout the brain and various organs. Therefore, it is possible that enkephalins influence mechanisms involved in lipid or glucose metabolism within these tissues [7,18]. Specifically, increased gluconeogenesis in the liver contributes to hyperglycemia in diabetes [31]. Under normal conditions, gluconeogenesis is suppressed in the liver when there is elevated blood glucose. However, the dysregulation of key regulatory enzymes, such as glucose-6 phosphatase or phosphoenolpyruvate carboxylase, may contribute to uncontrolled gluconeogenesis in the presence of hyperglycemia or decreased insulin levels; although in our study, we did not observe any difference in insulin levels between the two genotypes, we measured it a few days after the OGTT and ITT. As stated above, there are several studies utilizing both MOP and DOP knockout mice that show elevated liver enzymes, specifically those involved in fatty acid oxidation. The measured difference in these enzymes between knockout and wildtype mice likely influences the phenotype of these mice when fed an HFD [18,27]. Therefore, future studies need to be designed to measure the level and activity of enzymes, such as acetyl-CoA carboxylase (ACC-α) and peroxisome proliferator-activated receptor (PPAR-α). It is also necessary to further investigate the role of adipose tissue, as Brestoff et al. (2015) demonstrated that beiging through UCP-1 can be modulated by enkephalins [19]. Thus, future studies are also needed to determine the alteration in brown and white adipose tissues in the presence and absence of enkephalin to comprehensively understand the role of enkephalin in the regulation of glucose homeostasis.

## 5. Conclusions

The opioid system is relatively well known for regulating food consumption and energy homeostasis. In addition to the well-studied opioid receptors, enkephalins may also contribute to energy homeostasis. Our study adds to the current literature by demonstrating that glucose homeostasis is impaired in the absence of enkephalins, and this may be due to reduced insulin sensitivity. However, a great deal of work is needed to better understand the role of enkephalins in this process. We also need to determine the interaction of enkephalins with the neuropeptide systems controlling feeding behavior and glucose homeostasis. Investigating both the central and peripheral effects of enkephalin may reveal how enkephalins alter glucose homeostasis. Considering that we used mice lacking enkephalins throughout the body and throughout their life cycle, additional studies, using pharmacological and molecular biological tools, are needed to temporarily silence the enkephalin gene to have a better understanding of the role of this system in glucose homeostasis.

## Figures and Tables

**Figure 1 biomedicines-11-00671-f001:**
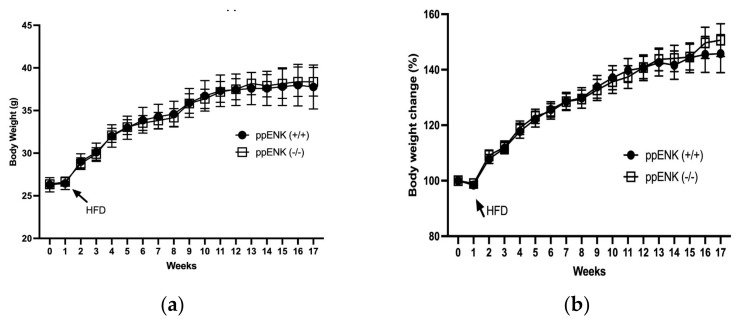
Body weight changes in mice lacking enkephalins (ppENK-/-) and their wildtype (ppENK+/+) littermates on HFD for 16 weeks. Data represent mean (±S.E.M.) of raw body weight (**a**) and percent of change in the original weight of mice of each genotype (**b**). The experiment was repeated four times with 5–7 mice per cohort (n = 21–23 mice per genotype). No significant difference was observed in either parameter between mice of the two genotypes (*p* > 0.05).

**Figure 2 biomedicines-11-00671-f002:**
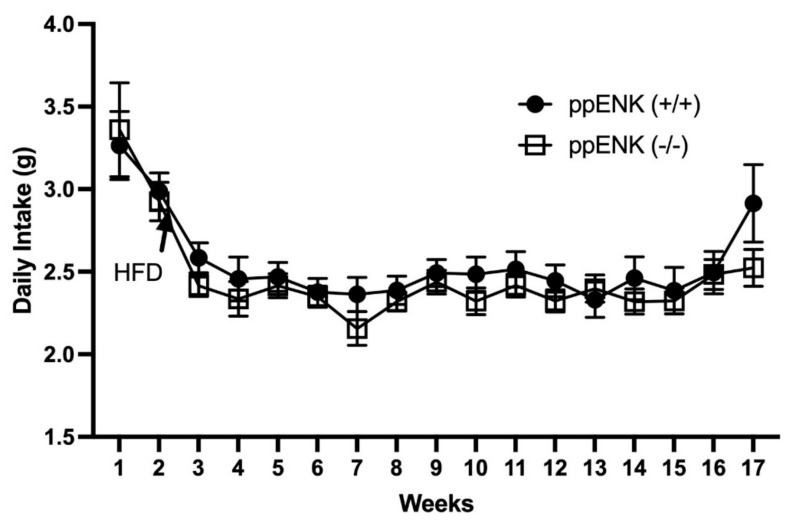
Changes in food intake in mice lacking enkephalins (ppENK-/-) and their wildtype (ppENK+/+) littermates when placed on HFD for 16 weeks. Data represent mean (±S.E.M.) of food intake (g) measured once a week for a total of 16 weeks in mice of the two genotypes. The experiment was repeated four times with 5–7 mice per cohort (n = 21–23 mice per genotype). No significant difference was observed in food intake between mice of the two genotypes (*p* > 0.05).

**Figure 3 biomedicines-11-00671-f003:**
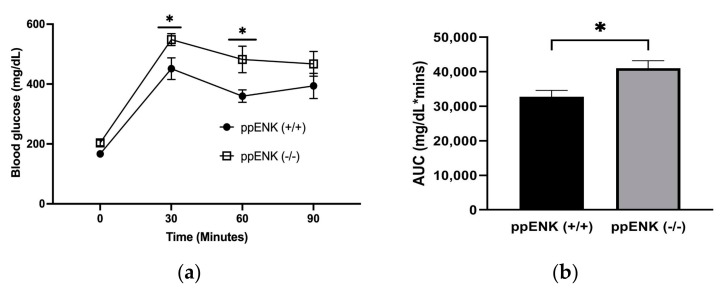
Oral glucose tolerance test (OGTT) performed after overnight fasting in mice lacking enkephalins (ppENK-/-) and their wildtype (ppENK+/+) littermates placed on HFD for 16 weeks. Data represent mean (±S.E.M.) of blood glucose levels (mg/dl) measured before and 30, 60, and 90 min after an oral glucose (2 g/kg, p.o.) challenge (**a**) and AUC (**b**). The experiment was repeated two times with 4–5 mice per cohort (n = 9 mice per genotype). * *p* < 0.05 a significant difference between mice of the two genotypes by Fisher’s LSD test (**a**) or unpaired student’s *t*-test (**b**).

**Figure 4 biomedicines-11-00671-f004:**
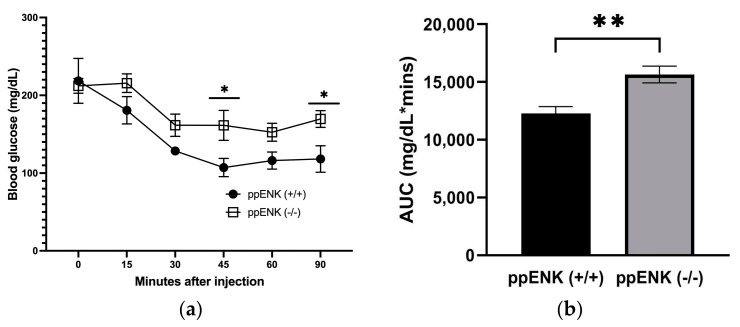
Insulin tolerance test (ITT) performed after six hours of fasting in mice lacking enkephalins (ppENK-/-) and their wildtype (ppENK+/+) littermates exposed to HFD for 16 weeks. Data represent mean (±S.E.M.) of blood glucose levels (mg/dL) measured before and 30, 60, and 90 min after an insulin (0.75 g/kg, i.p.) challenge (**a**) and calculated AUC (**b**) in wildtype and knockout mice (n = 5 mice per genotype). A significant difference between mice of the two genotypes by the Fisher’s LSD test (* *p* < 0.05, panel (**a**)) or unpaired student’s *t*-test (** *p* < 0.01, panel (**b**)).

**Figure 5 biomedicines-11-00671-f005:**
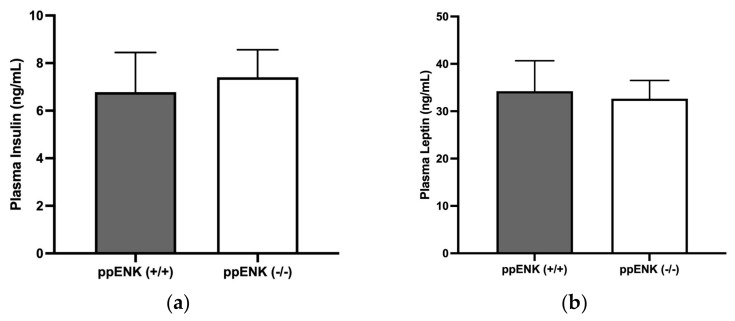
Plasma levels of insulin (**a**) and leptin (**b**) in mice lacking enkephalins (ppENK-/-) and their wildtype (ppENK+/+) littermates on HFD for 16 weeks. Data represent mean (±S.E.M.) of plasma levels of insulin ((**a**), 8–10 mice per genotype) and leptin ((**b**), n = 9–11 mice per genotype) in ng/mL. No significant difference was observed in either parameter between mice of the two genotypes (*p* > 0.05).

## Data Availability

Data will be made available upon reasonable request.

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
