# Peer review of "The Involvement of Endogenous Enkephalins in Glucose Homeostasis†"

_biomedicines, 2023, doi:10.3390/biomedicines11030671_

Round 1

Reviewer 1 Report

Escolero and colleagues prepared a manuscript where they pretend to evaluate the role of endogenous enkephalins in energy and glucose homeostasis. Understanding the mechanisms and identifying novel players involved in the development of obesity and insulin resistance, which will ultimately lead to other pathologies such as diabetes, is of highly relevance. For that reason, elucidating the impact of knocking-down enkephalin in glucose metabolism is very timely and important.

Nevertheless, the information given in the current manuscript seems to be a project in its very beginning and should to be deepened. Otherwise, its interested will be very limited. The results from the oGTT and ipITT are very exciting, but other than that the article is relatively poor.

It looks like the methodology was not very well projected. For instance, mice are social animals and should not be in individual cages.

The value of Figure 5 is limited, because when blood is collected for measuring insulin (and leptin), mice should either be fasted (as discussed by the authors) or reefed for a determined amount of hours. However, to determine if the difference was related to the insulin release from the pancreas, a glucose-stimulated insulin secretion (GSIS) test should rather be performed.

It is not clear why the mice were sacrificed, as blood can be easily, and relatively painless, collect in live (or minimal anesthetized) mice. Nevertheless, if mice were sacrificed, tissues must have been collected, so why not include some gene expression data on metabolic tissues, such as liver, adipose tissues and skeletal muscle?

As for analysis presented, I would recommend determining the areas under the curve (AUC) for both GTT and ITT.

The exact number of mice for each time point should be indicated in the legend.

In figure 3, it seems there is no error bar at fasting (time 0). If so small that can't be seen, wasn't the fasting glycemia significantly different?

The results section is not attractive, as it only directs the reader to the figure, without truly exploiting the goal of the experiment, nor the conclusions of each result.

Some demonstration that these mice are truly KO for enkephalin should be included.

The authors should not use the term "glycemia control" when referring to "glucose intolerance", as those are distinct features.

The data on the enkephalin knockout mice fed a chow diet should be shown.

Author Response

Dear Reviewer,

I would like to thank you for the time to review our manuscript. We are also indebted to you for your constructive and insightful comment. We modified our manuscript according to your comments, but we are willing to further revise our manuscript if there are any additional issues.

Sincerely,

Kabirullah Lutfy, Ph.D.

Reviewer 2 Report

The authors submitted a research article in which they elucidated potential role of the opioid system, specifically, enkephalin in the regulation of glucose homeostasis and in the pathophysiology of type 2 diabetes mellitus. Although the findings of the review seem to be impressive, I would like to put forward several comments to discuss.

1. Please, transform the term ' type II diabetes' into 'type 2 diabetes mellitus'.

2. Section Introduction does not open a purpose of the review and in fact seems to be superficial. Please, concentrate on the initial hypothesis of the review, but not hystorical references. Add please, clear statistical report regarding the problem, indicate a tendency in a prevalence, provide challanging and add your arguments why he problem appears to be important for readers.

3. Methods: it retained to be unclear why leptin was used to the list of tests. What's about other adipocytokines ? Blood sampling should be thoroughly described. Statistics should be improved. Whether multiple comparissons were doe and what tests were used. Authors used e Fisher’s LSD post-hoc test. The aim of the article ('the role of enkephalins in energy and glucose homeostasis') requires more advance statistics that used to. Please, consider regression analysis with further reclassification of factors with AUCs and/or INR, IDI.

4. Conclusion parz seems to show rather general idea than close findings. Please, provide more readable conclusive part.

Author Response

(The authors gave the same response as above.)

Reviewer 3 Report

The manuscript entitled "The involvement of endogenous enkephalins in energy and 2 glucose homeostasis" written by Escolero V. et al explores an interesting topic. Although the deletion of enkephalins did not affect food intake or body weight, it was proven that it was involved in glucose homeostasis. Overall, the manuscript is well written, the study design is appropriate and the methods are reproductible, the results are clearly presented, and discussions are well conducted. 

However, the following major comments should be addressed:

1. Does "†" should be mentioned in the title or could it be added somewhere else in the manuscript?

2. The Abstract contain no data regarding the number of used animals, or the study results. The phrase containing year 1975 could be replaced by a phrase containing newer evidences. 

3. The number of keyword is higher than the usual number of 5 recommended. 

4. The first paragraph of Introduction chapter contains data from the journal manuscript template. 

5. References regarding obesity prevalence could be updated to the newest available data. 

6. The correct term is "diabetes type 2", in particularly mentioned by American Diabetes Association. 

7. The phrase "We utilized enkephalin knockout mice 93 that were previously generated by Konig and colleagues [16]" should be moved to Material and Methods section. 

8. Did the authors used a protocol(s) for feeding, performing OGTT,  Intraperitoneal Insulin Tolerance Test? Please, provide the references. 

9. The Conclusion chapter contains references from the literature that should be placed in the Discussion chapter. 

Author Response

(The authors gave the same response as above.)

Round 2

Reviewer 1 Report

The authors have provided a revised version of the manuscript and have minimally answered to my concerns. I recommend removing the word "energy" from the title, and also to aim to a Communication format and not an Article.

Reviewer 2 Report

The authors submitted a revised version of the paper along with clear explanation of the ways by which the corrections were made. I have no serious concenrns about the paper in its revised version.

Reviewer 3 Report

The quality of the manuscript has improved after answering the addressed comments, except for the 2nd one. The authors made additional changes, not previously requested. The text is difficult to read given the multiple corrections and interventions. 

What does the word "retareded" refer to? 

Why is italics font used starting line 243? 
